# Coverage of the national surveillance system for human *Salmonella* infections, Belgium, 2016-2020

Nina Van Goethem[1,2]*, An Van Den Bossche[3], Pieter-Jan Ceyssens[3], Adrien Lajot[1], Wim Coucke[4], Kris Vernelen[4], Nancy H. C. Roosens[5], Sigrid C. J. De Keersmaecker[5], Dieter Van Cauteren[1‡], Wesley Mattheus[3‡]

**1** Scientific Directorate of Epidemiology and Public Health, Sciensano, Brussels, Belgium, **2** Department of Epidemiology and Biostatistics, Institut de Recherche Expérimentale et Clinique, Faculty of Public Health, Université Catholique de Louvain, Woluwe-Saint-Lambert, Belgium, **3** National Reference Centre for Salmonella and Shigella, Sciensano, Brussels, Belgium, **4** Quality of laboratories, Sciensano, Brussels, Belgium, **5** Transversal Activities in Applied Genomics, Sciensano, Brussels, Belgium

‡ These authors contributed equally to this work (Joint Senior Authors).
* nina.vangoethem@sciensano.be

**Data Availability Statement:** All relevant data are within the manuscript and its Supporting Information files.

**Funding:** This work has been funded by Sciensano, the Belgian institute for health, through the

## Abstract

### Introduction

The surveillance of human salmonellosis in Belgium is dependent on the referral of human *Salmonella* isolates to the National Reference Center (NRC). Knowledge of current diagnostic practices and the coverage of the national *Salmonella* surveillance system are important to correctly interpret surveillance data and trends over time, to estimate the true burden of salmonellosis in Belgium, and to evaluate the appropriateness of implementing whole-genome sequencing (WGS) at this central level.

### Methods

The coverage of the NRC was defined as the proportion of all diagnosed human *Salmonella* cases in Belgium reported to the NRC and was assessed for 2019 via a survey among all licensed Belgian medical laboratories in 2019, and for 2016–2020 via a capture-recapture study using the Sentinel Network of Laboratories (SNL) as the external source. In addition, the survey was used to assess the impact of the implementation of culture-independent diagnostic tests (CIDTs) at the level of peripheral laboratory sites, as a potential threat to national public health surveillance programs.

### Results

The coverage of the NRC surveillance system was estimated to be 83% and 85%, based on the results of the survey and on the two-source capture-recapture study, respectively. Further, the results of the survey indicated a limited use of CIDTs by peripheral laboratories in 2019.

BeREADY project and was conducted in the framework of the Adonis project, supported by funding from the European Union's Horizon 2020 Research and Innovation programme under grant agreement No 773830: One Health European Joint Programme.

**Competing interests:** The authors have declared that no competing interests exist.

## Conclusion

Given the high coverage and the limited impact of CIDTs on the referral of isolates, we may conclude that the NRC can confidently monitor the epidemiological situation and identify outbreaks throughout the country. These findings may guide the decision to implement WGS at the level of the NRC and may improve estimates of the true burden of salmonellosis in Belgium.

## Introduction

Salmonellosis is the second most frequently reported bacterial gastrointestinal infection in humans in Europe, despite a significantly long-term decreasing trend in human cases since 2008 [1, 2]. In 2019, a total of 87,923 confirmed cases of salmonellosis in humans were reported in the EU and *Salmonella* was responsible for 17.9% of all food-borne outbreaks during 2019 [1].

The surveillance of human salmonellosis is in general based on human *Salmonella* isolates that are sent to a reference laboratory for confirmation, further typing and antibiotic susceptibility testing [3]. Over the last decade, whole genome sequencing (WGS) is increasingly being implemented at the level of reference microbiology centers and replaces a series of traditional typing methods for isolates [4–6]. Full potential of such high-resolution typing can be achieved when applied to a representative national data set in order to obtain a comprehensive view on circulating strains and enable the linkage of outbreak cases from distinct geographical regions and/or with potential food sources or contaminants. However, a centralized approach is highly dependent on the referral of isolates by frontline laboratories. A potential threat to national public health surveillance programs is the recent implementation of culture-independent diagnostic tests (CIDTs) at the level of hospitals and clinical laboratories for the detection of enteric pathogens to guide clinical decision making, potentially resulting in less submission of culture isolates to reference centers [7, 8].

Laboratory surveillance of human *Salmonella* cases in Belgium is based on the reporting by the National Reference Center (NRC). The NRC receives human *Salmonella* isolates that are voluntarily sent by peripheral clinical microbiology laboratories for confirmation, antibiotic susceptibility testing and further sero- and subtyping. Molecular subtyping by multiple-locus variable number tandem repeat analysis (MLVA) is routinely performed for the two most important serotypes (*S. enterica* subsp. *enterica* serovar Enteritidis and Typhimurium). WGS is applied in specific cases involving multidrug-resistant, invasive or outbreak-related strains. The NRC ensures national epidemiological surveillance of human *Salmonella* infections, by prospectively identifying outbreaks, and by following the long-term spatial and temporal trends. Other complementary surveillance systems exist, such as a sentinel network of laboratories (SNL) [9]. It consists of a limited number of laboratories ('sentinel') that are distributed evenly in Belgium and that weekly reports the number of isolated *Salmonella* spp. to the Belgian Scientific Institute of Health, Sciensano.

Usually, surveillance systems do not capture all cases within a population due to the under-ascertainment of cases (i.e. the number of infected individuals who are not diagnosed) and the under-reporting of the diagnosed cases, resulting in a gap between the real and registered incidence of human salmonellosis [10]. Knowledge of current diagnostic practice and the coverage of the national *Salmonella* surveillance system in Belgium, defined as the proportion of all isolates from diagnosed cases with a laboratory confirmation that are sent to the NRC, is

important to correctly interpret surveillance data and trends over time. Changes in laboratory practice such as the use of CIDTs may impact the current surveillance system that relies upon culture confirmation and the referral of isolates.

In this study we aim to evaluate current diagnostic practices and assess the coverage of the NRC surveillance system for *Salmonella* based on a survey that was conducted among licensed Belgian medical laboratories in 2019, as well as via a capture-recapture study over the 2016–2020 time period using the SNL as the external source. Subsequently, these results will contribute to estimate the true burden of salmonellosis in Belgium and to evaluate the appropriateness of implementing WGS at the central level of the NRC.

## Materials and methods

### *Salmonella* epidemiology in Belgium based on the NRC and the SNL

A 'case' is defined as an individual that meets the confirmed laboratory criteria for diagnosis, i.e. isolation of *Salmonella* spp. from a clinical specimen. A case is not counted as a new case if laboratory test results were reported within 90 days of a previously reported infection in the same individual. The official national figures for human salmonellosis are based on the number of *Salmonella* isolates from unique cases that are sent to the NRC by peripheral laboratories. The isolates are accompanied by a form with epidemiological information that includes the age, gender, and postal code of the patient, together with the associated clinical picture and recent travel history. The trend of the annual incidence rate (reported cases/100,000 inhabitants) is presented per region (Flanders, Wallonia and Brussels) and per serotype for the 2016–2020 period. The two most common serotypes in Belgium, *Salmonella* ser. Typhimurium (including Typhimurium-like strains) and *Salmonella* ser. Enteritidis, as well as the other serotypes together are presented. Next, the national trend for the 2016–2020 period based on the NRC is compared to the number of cases as reported through the SNL. The SNL database contains the cases that are reported to Sciensano by laboratories participating to the sentinel network (i.e. a selection of medical laboratories in Belgium). The number of medical laboratories participating in the sentinel network ranged between 38 and 47 laboratories from 2016 to 2020. The transferred variables include some patient's demographic data allowing the identification of duplicates, i.e. date of birth, gender and postal code [9]. The NRC and SNL data accessed in the context of the present study had not been collected for research purposes but as part of the routine data collection for epidemiological surveillance, as stated in the Public Register dated 25/04/1997 in accordance with article 18 of the law of 08/12/1992 of the Belgian Government regarding the protection of the privacy of the individual when dealing with personal data. The aforementioned registration in accordance with the Belgian Privacy Commission stipulates in its §9 that no written informed consent from the patients is required for the collection and analysis of epidemiological data and treatment success collected for a public health purpose.

### Laboratory survey

The survey was linked to the External Quality Assessment (EQA) for medical laboratories conducted by the service Quality of laboratories from Sciensano to assess the quality of laboratory analyses by sending an unknown isolate with clinical context to all participating laboratories. Participation to EQA is mandatory for all licensed Belgian medical laboratories in order to be recognized by the national health insurance reimbursement system (RIZIV-INAMI) as described under the Belgian law. The EQA was performed in January 2020 and an isolate *Salmonella enterica* serovar Poona from a patient who was part of a European outbreak associated with infantile milk powder in 2019 [11] was sent to all 121 licensed laboratories performing bacteriology in Belgium. The additional survey consisted of a short questionnaire about the

number of cases diagnosed in 2019, the techniques used for diagnosis, and the reasons why isolates were sent to the NRC. The laboratory survey data did not involve personal data as only aggregated data per laboratory was collected.

## Reimbursement data

Data from the RIZIV-INAMI were used to obtain the number of reimbursed stool cultures per health insurance identification number. These data were used to calculate the *Salmonella* positivity rate and to subsequently estimate the total number of cases in Belgium in 2019 taking into account the licensed laboratories that did not participate in the survey. The reimbursement data did not involve personal data as only aggregated data per laboratory was collected.

## Coverage NRC based on the laboratory survey

The coverage of the NRC surveillance was assessed by comparing the number of isolates from unique cases with an isolation date in 2019 sent to the NRC by the peripheral laboratory sites to the total number of cases diagnosed by the licensed medical laboratories in Belgium as reported in the survey. The peripheral laboratory sites that sent isolates to the NRC were linked to the licensed medical laboratories that participated in the survey, using the health insurance identification number to estimate the coverage by laboratory. A laboratory can have >100% coverage when they report fewer cases in the survey than the number of isolates from unique cases sent to the NRC.

## Capture-recapture

An alternative approach to assess the coverage of surveillance is via a 'capture-recapture' study [10] that consists of comparing at least two independent data sources that provide surveillance information on the same disease to estimate or adjust for the extent of incomplete ascertainment [12]. By matching several sources of information originating from the same population, this method allows to estimate the 'true' total number of cases $\widehat{N}$ and the number of cases not identified by any of the sources $\widehat{x}$ (i.e. 'missed' cases) [10, 12–14]. The estimated total number of cases can be calculated based on the Chapman's formula [15] for two-source capture-recapture studies $\widehat{N} = \frac{(n_1+1)(n_2+1)}{m+1} - 1$, where $n_i$ is the number observed in source $i$ and $m$ the number observed in both sources (i.e. 'matched' cases) [16]. The 95% confidence interval for the estimated total number of cases was calculated using the Chao's lower-bound estimator [10, 16, 17]. External completeness (here denoted as the coverage), i.e. the proportion of 'true' cases detected by a source, is then calculated by dividing the number of cases reported to surveillance $n_i$ by the estimated 'true' total number of cases $\widehat{N}$ in the study population. A Wald interval was used to estimate the confidence interval of the proportion. The number of cases which could have been missed by both sources is estimated by $\widehat{x} = \widehat{N} - (n_1 + n_2 - m)$.

This methodology was used to estimate the coverage of the NRC database for the time period 2016–2020, using the SNL surveillance system as the external source. All reported cases within the NRC and SNL databases were de-duplicated (based on age, gender, postal code) within a 90-day interval. Cases from both data sources were matched per year based on date of birth, gender, and postal code. Analyses were done in R 3.6.3. [18].

## Results

### Salmonella epidemiology in Belgium

The evolution of the number of cases (S1 Table) per 100 000 inhabitants annually reported to the NRC between 2016 and 2020 is shown for *Salmonella* ser. Typhimurium in Fig 1A, for

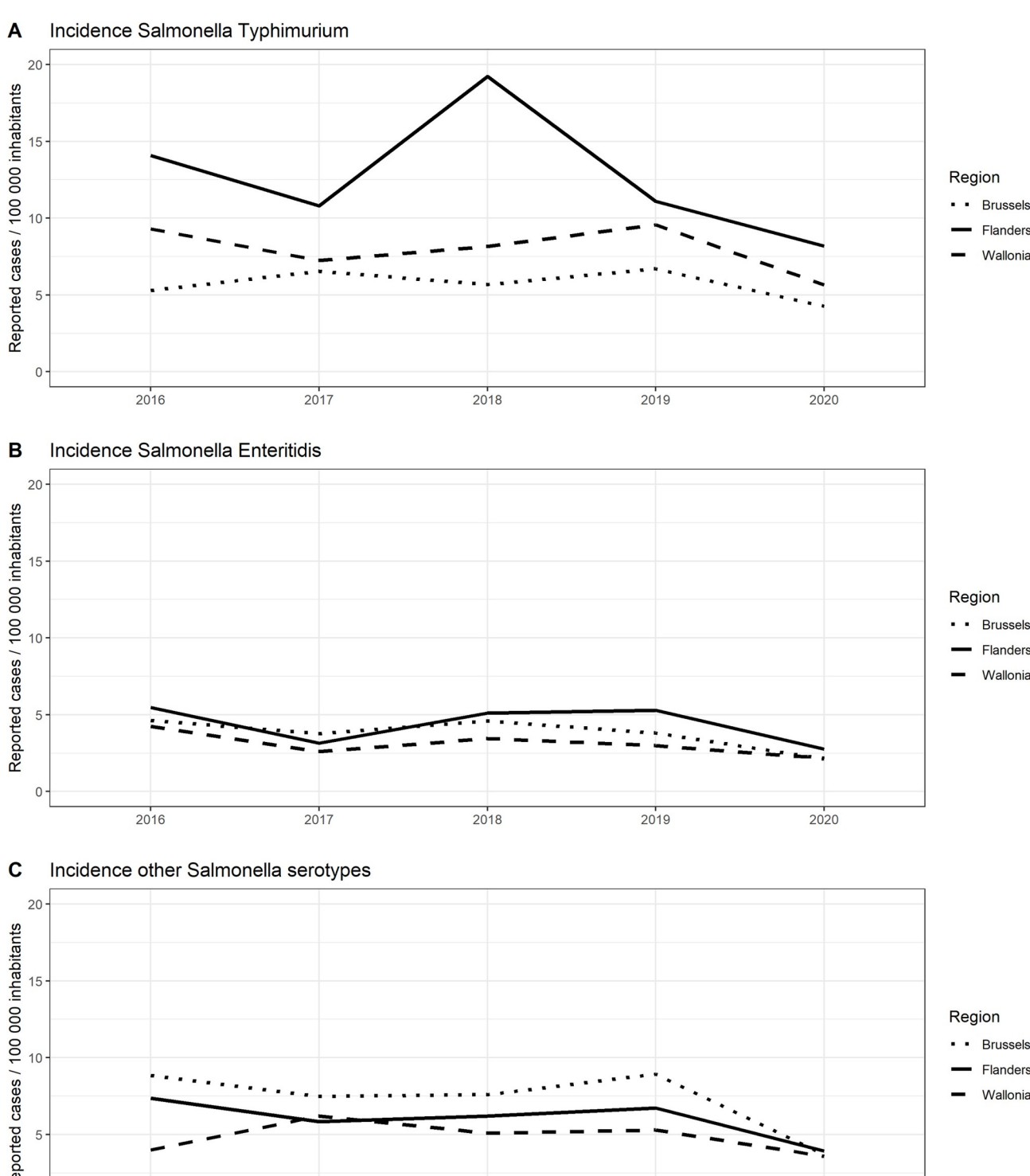

**Fig 1. Number of *Salmonella* cases per 100 000 inhabitants reported by the National Reference Center (NRC) per region, Belgium, 2016–2020.**
Panel A: *Salmonella* ser. Typhimurium; Panel B: *Salmonella* ser. Enteritidis; Panel C: other *Salmonella* serotypes (panel C).

*Salmonella* ser. Enteritidis in Fig 1B, and for the other *Salmonella* serotypes in Fig 1C. For *Salmonella* ser. Typhimurium there was an increase in 2018 compared to 2017 because of a large outbreak in Flanders.

Fig 2 (underlying numbers presented in S2 Table) compares the annual number of cases reported by the NRC and the SNL for the 2016–2020 period. The annual trends observed between both surveillance systems are similar. The lower numbers reported by the SNL are expected given the limited number of laboratories participating.

## Coverage of the NRC surveillance system assessed through a laboratory survey

There exists no unique mapping system for all laboratory sites in Belgium, resulting in different administrative aggregations used by the NRC database, EQA survey and the reimbursement data. However, the individual laboratories that occur within the different sources can be linked to each other as presented in Fig 3. The survey results are presented in S3 Table. A total of 121 licensed laboratories performing bacteriology were identified for the EQA in January 2020 and 116 participated to the survey, resulting in a participation rate of 96%. From the 116

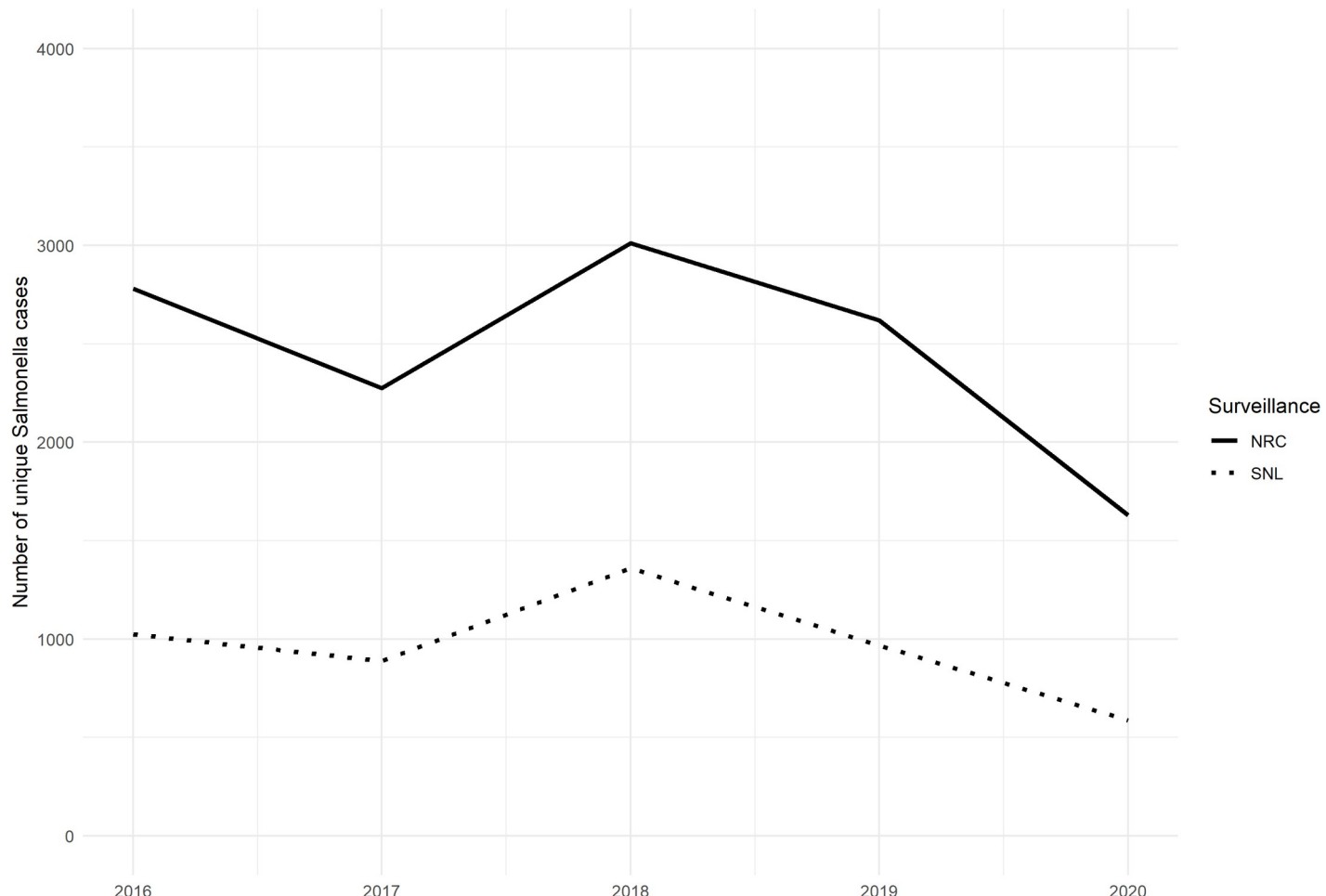

**Fig 2. Number of *Salmonella* cases in the National Reference Center (NRC) database (solid line) and the Sentinel Network of Laboratories (SNL) database (dotted line), Belgium, 2016–2020.**

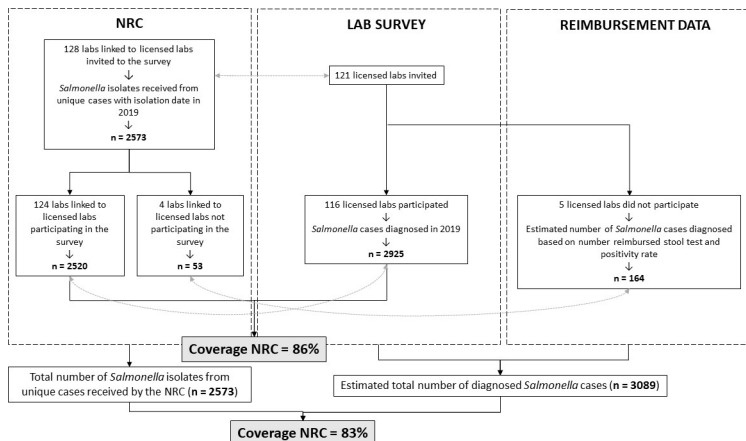

**Fig 3. Linking peripheral laboratory sites that sent isolates to the National Reference Center (NRC) with licensed medical laboratories that were invited to the survey to estimate the coverage of the NRC surveillance system, Belgium, 2019.**

licensed laboratories that have participated, three were grouped together (different activity centers of one central clinical laboratory) and therefore a total of 113 questionnaires were returned. In total, 2925 cases in 2019 were reported in the survey by the 116 licensed laboratories.

The NRC received 2573 isolates from unique cases from 128 peripheral laboratory sites that could be linked to the 121 licensed laboratories invited to the survey. Among all licensed laboratories that were invited to the survey, only two did not send isolates to the NRC in 2019. A total of 124 peripheral laboratory sites from the NRC database could be linked to the 116 licensed laboratories that participated to the survey, responsible for 2520 unique cases in the NRC database. This corresponds to a coverage of 86% (95% CI: 85–87%) of the NRC surveillance system.

A positivity rate of 0.72% was obtained for the participating laboratories by dividing the number of cases as reported in the survey by their number of reimbursed stool cultures (reimbursement data presented in S4 Table). Subsequently, the number of cases was estimated for the five non-participating laboratories based on this positivity rate and their number of reimbursed stool cultures. When taking into account the estimated number of cases from the five non-participating licensed laboratories based on the reimbursement data (n = 164) in addition to number of cases as reported through the survey (n = 2925), the total number of cases in Belgium in 2019 was estimated to be 3089. By comparing this to the total number of isolates from unique cases received at the NRC in 2019 (n = 2573), the coverage of the NRC surveillance was estimated to be 83% (95% CI: 82–85%).

The majority of laboratories (64%) had a coverage between 80–120%, 28% had a coverage below 80%, and 8% had a coverage above 120%. The latter were laboratories that reported only a small number of cases. A total of 18 laboratories had a coverage greater than 100% (because they reported fewer cases in the survey than the number reported to the NRC). The median coverage of the NRC database with respect to the licensed laboratories that participated in the survey is 91% (IQR: 75–100%). There was no significant difference between regions in the coverage per activity center (p = 0.194) (Fig 4).

Concerning the method of diagnosis, 70% (79/113) indicated they used culture, 42% (47/113) performed the combination of culture and serotyping, and 70% (79/113) indicated they used MALDI-TOF. Only 4% (5/113) of laboratories used multiplex PCR (i.e. a CIDT) to diagnose

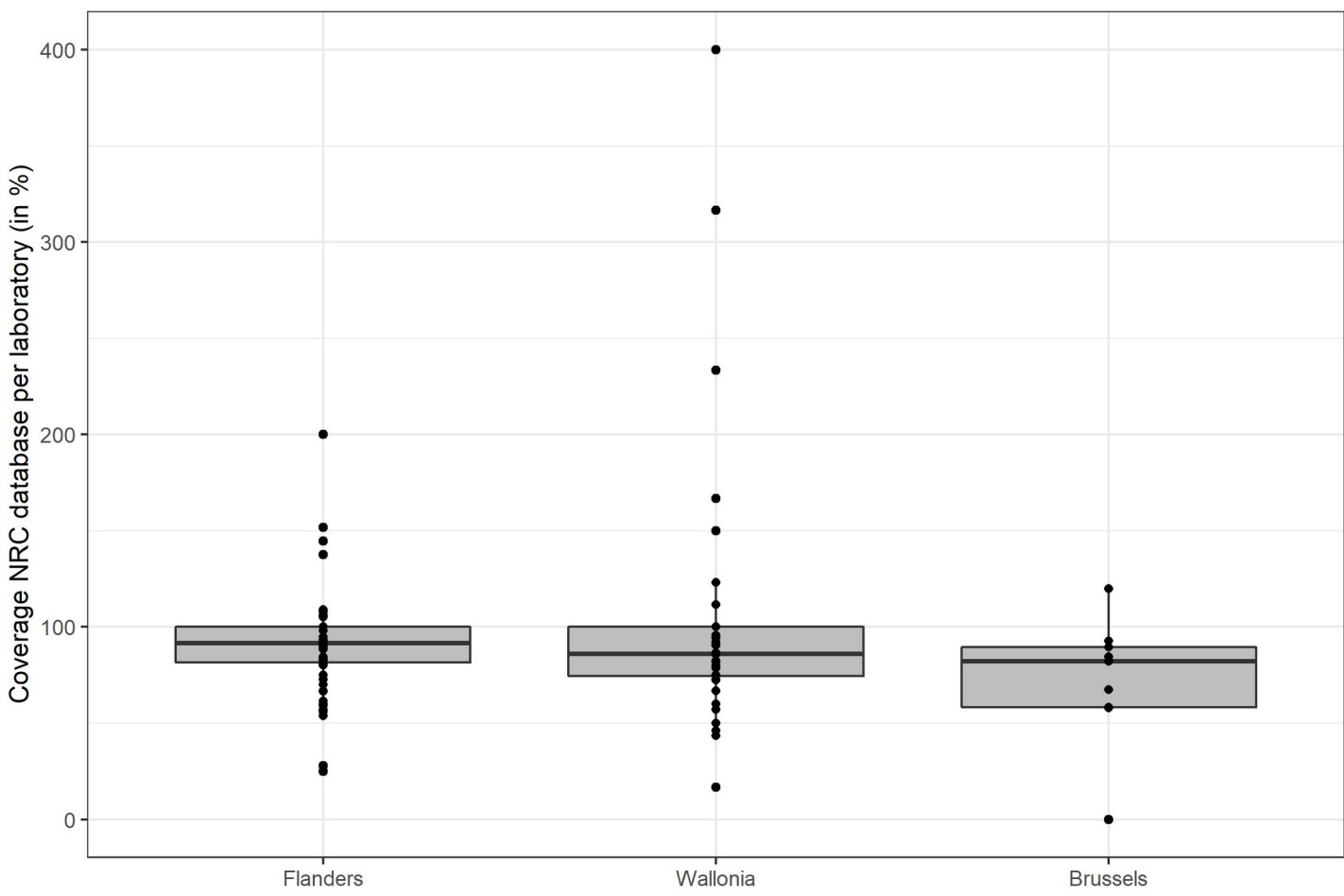

**Fig 4. Coverage of the NRC surveillance system per licensed laboratory, per region, Belgium, 2019.**

*Salmonella* cases in 2019. All laboratories except one indicated they sent all serotypes to the NRC. The main reasons for sending isolates to the NRC were for epidemiological reasons (n = 91), for confirmation and/or antibiotic resistance (n = 69), and for further serotyping (n = 9).

## Coverage of the NRC surveillance system assessed through a capture-recapture study

A total of 12,316 and 4,822 cases were reported in the NRC and SNL databases, respectively, from 2016 to 2020. After removing cases with missing date of birth, gender, or postal code, a total of 11,017 and 4,548 cases from the NRC and SNL database respectively, were available for the capture-recapture analysis. A total of 3,869 cases could be matched based on date of birth, gender, and postal code (Table 1). Using the Chapman equation [15] for capture-recapture [12], it was estimated that there were a total number of 12,950 (95% CI: 12,823–13,077) expected cases over the 2016–2020 period (Table 1). Based on this, the estimated number of missed cases over this period was 1,254. The coverage of the NRC network and the SNL network were estimated to be 85% (11,017/12,950) (95% CI: 84–86%) and 35% (4,548/12,950) (95% CI: 34–36%) respectively. When taking into account the cases with missing date of birth, gender, or postal code, the estimated total number of cases was 13,350 (95% CI: 15,172–15,528) and the coverage of the NRC and SNL network were estimated to be 80% (12,316/

**Table 1. A two-source capture-recapture analysis to estimate the 'true' number of *Salmonella* cases in Belgium, 2016–2020.**

| SNL | NRC | | |
| --- | --- | --- | --- |
| | Identified | Not identified | Total |
| Identified | 3869 | 679 | 4548 |
| Not identified | 7148 | *1254* | *8402* |
| Total | 11017 | *1933* | *12950* |

NRC: National Reference Center; SNL: Sentinel Network of Laboratories. In italics: numbers estimated by capture-recapture using Chapman's formula [15].

15,350) (95% CI: 80–81%) and 31% (4,822/15,350) (95% CI: 31–32%), respectively. The numbers underlying the capture-recaptures analysis are presented in S5 Table.

As trends between both data sources were similar (Fig 2), the coverage of the NRC database remained stable for the 2016–2020 time period (Fig 5).

## Discussion

The coverage of the national *Salmonella* surveillance system in Belgium by the NRC was estimated by two methods, an individual laboratory survey and a capture-recapture study, which

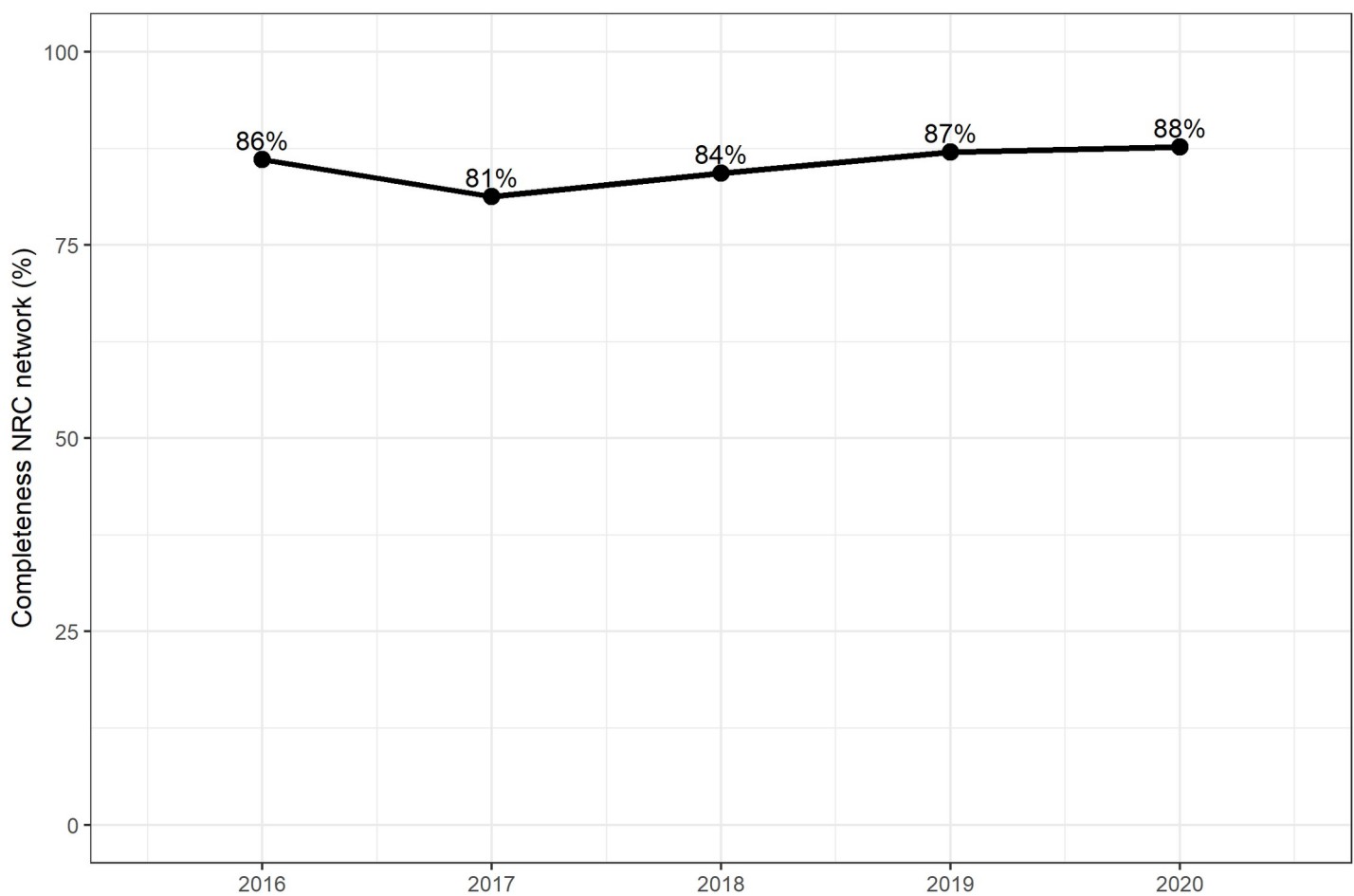

**Fig 5. Coverage of the NRC surveillance system as estimated through a capture-recapture study, 2016–2020, Belgium.**

resulted in an estimated coverage of 83% and 85%, respectively. These coverages are higher than reported in other European countries such as France (48%) and The Netherlands (64%) [1]. The NRC is a well-established recognized reference laboratory for more than 30 years in Belgium. Besides *Salmonella*, this NRC is also the reference lab for *Shigella*, Yersinia (*Y. entero-colitica* and *Y. pseudotuberculosis*), *Listeria monocytogenes*, *Mycobacterium Tuberculosis*, and *Neisseria meningitidis*. The latter three are severe infections with mandatory notification towards regional health inspection services and for which sending of isolates to the NRC is strongly encouraged in order to detect clusters and implement preventive measures. The habit and reflex of all laboratories to send isolates of these severe infections covered by mandatory notification may contribute to a better coverage of less severe infections such as salmonellosis (and probably also shigellosis and yersiniosis) for which sending isolates is voluntary, but towards the same recognized national reference laboratory.

The capture-recapture study provided evidence that the coverage of the NRC surveillance network remained stable over the past 5 years. Even in 2020, when there was a strong decrease in (diagnosed) *Salmonella* cases probably related to the impact of the COVID-19 crisis, the coverage of the NRC surveillance remained high (88%). As a limitation of the capture-recap-ture study, we acknowledge that potentially not all underlying assumptions to perform such a study were met and these violations may impact the reliability of the estimate [12]. These assumptions are (1) identification of all and only true common cases, (2) the population under study must be 'closed' cohort with no losses or new entries in a given time period, (3) each case must have the same probability of being identified by each source (also known as homogeneity) and (4) the two sources should be independent. Case homogeneity and the independence of sources are difficult assumptions to verify and in reality are unlikely to be fully met within a healthcare setting [13].

The survey, on the other hand, benefited from the fact that participation to EQA is mandatory for all licensed Belgian medical laboratories in order to be recognized and reimbursed. As a consequence, almost all active laboratories in January 2020 returned the additional questionnaire resulting in a high participation rate of 96%. Data on reimbursed stool samples per licensed laboratory was used to account for the non-participating laboratories and allowed us to confidently estimate the total number of *Salmonella* diagnoses in 2019. The coverage of the NRC surveillance, based on the results of the survey in combination with the reimbursement data, was estimated to be 83% in 2019. This is in line with the estimated coverage of the NRC surveillance of 85% for the 2016–2020 period based on the capture-recapture study. The consistent results between both methods support the future use of a capture-recapture analysis using the SNL as the external sources to follow-up the coverage of the NRC surveillance system over time.

The survey and the NRC surveillance are completely independent data sources and apparently some laboratories, mainly those with only few *Salmonella* diagnoses, reported less cases in the survey as compared to their number of cases in the NRC database resulting in a coverage above 100%. However for the majority of laboratories the number of cases in the NRC database was close to their reported number of cases in the survey, resulting in a median coverage of 91% (IQR: 75–100%).

The survey revealed that laboratories do not make a selection based on the serotype to send isolates to the NRC. No regional differences in laboratory practice were observed that may explain the higher incidence of *Salmonella* ser. Typhimurium in Flanders (whereas this is not the case for *Salmonella* ser. Enteritidis or the other serotypes). Therefore, we are confident that the observed difference in incidence in serotypes between the different regions reflects the reality and is not due to a selective sending of isolates. Possible explanations of a higher incidence of *Salmonella* ser. Typhimurium in Flanders as compared to the other regions could be

differences in food consumption patterns and/or a higher environmental spread due to abundant pig farms in Flanders. However, these hypotheses require further investigation by combining data from food, animals and humans in a One Health perspective.

The answers on the question in the survey regarding the method of diagnosis indicate that the use of CIDTs for the identification of *Salmonella* is rather limited in January 2020 in Belgium. However, the implementation of CIDTs and the impact on microbiological surveillance should be regularly monitored to detect possible changes in laboratory practice, for example related to the COVID-19 epidemic. Although CIDTs have the potential to improve estimates of disease burden given their high sensitivity and relative ease of use, the routine implementation of CIDTs in public health surveillance may challenge the interpretation of trends in disease incidence [19]. Moreover, referral of isolates to reference centers is still needed for strain characterization and antimicrobial resistance testing. In addition to the use of CIDTs, the ongoing reorganization of clinical laboratories in bigger platforms that cover several distal sites may impact laboratory practice and referral of isolates to the NRC.

The high coverage of the NRC surveillance system advocates for the implementation of WGS at this central level applied to the set of isolates received from the peripheral laboratories in Belgium. WGS generates highly discriminatory data on *Salmonella* isolates and enables the detection of outbreaks at an earlier stage and a more efficient surveillance of sporadic cases when integrated in a routine and real-time surveillance system [20]. Routine sequencing of the isolates received by the NRC would result in a representative database of circulating strains that enables the estimation of background diversity in the population and a better discrimination of new outbreak isolates [6]. From an economic perspective, Alleweldt *et al* [21] showed that only 0.2–1.1% of reported salmonellosis cases would need to be avoided each year through the use of WGS (e.g. by earlier outbreak detection) in order to make its implementation cost-neutral from a public health perspective. In addition, reference laboratories that deal with a high volume of samples achieve a lower per-sample cost than smaller institutions processing a lower amount of samples and the implementation at a central level would alleviate infrastructure investments and staffing requirements [21, 22].

Finally, this study can help to improve estimates of the true burden of salmonellosis in Belgium. Indeed, in addition to case under-ascertainment, i.e. the number of infected individuals who do not seek healthcare, the degree of underestimation and the sensitivity of a surveillance system is affected by the coverage of the system and the subsequent underreporting, i.e. the number of diagnosed patients that are not reported to the surveillance system [10, 23]. The results of this study will contribute to a more accurate and precise estimate of the true burden of salmonellosis in Belgium [24].

## Conclusions

Given the high coverage and the limited impact of CIDTs on the referral of isolates, we may conclude that the surveillance based on the NRC approximates the true number of laboratory confirmed *Salmonella* cases in Belgium allowing to monitor circulating strains and identify outbreaks throughout the country. The results of this study will also contribute to a more precise and accurate estimate of the true burden of salmonellosis in Belgium and provides arguments for the implementation of WGS at this central level.

## Supporting information

**S1 Table. Number of *Salmonella* cases per year, per serotype and per region.**
(XLSX)

**S2 Table. Number of *Salmonella* cases in the in the National Reference Center (NRC) database and the Sentinel Network of Laboratories (SNL) database.**
(XLSX)

**S3 Table. Laboratory survey data.** Aggregated data per laboratory (anonymized) on the number of *Salmonella* cases reported in the survey, the number of *Salmonella* isolates from unique cases sent to the National Reference Center (NRC), the method of diagnosis used, the reasons for sending isolates to the NRC, and the serotypes sent to the NRC.
(XLSX)

**S4 Table. Stool reimbursement data.** Aggregated data per laboratory health insurance reference number (anonymized) on the number of reimbursed stool cultures and the number of *Salmonella* cases reported in the survey.
(XLSX)

**S5 Table. Capture-recapture analysis.** Numbers per year used to calculate the coverage of the National Reference Center (NRC) and Sentinel Network of Laboratories (SNL) through a capture-recapture analysis.
(XLSX)

## Acknowledgments

We would like to thank all people in charge of the sentinel and peripheral laboratories for their collaboration, their regular transfer of isolates and/or data, and their participation in the EQA survey.

## Author Contributions

**Conceptualization:** Nina Van Goethem, Dieter Van Cauteren, Wesley Mattheus.

**Data curation:** An Van Den Bossche, Adrien Lajot, Wim Coucke, Kris Vernelen, Dieter Van Cauteren, Wesley Mattheus.

**Formal analysis:** Nina Van Goethem.

**Funding acquisition:** Pieter-Jan Ceyssens, Nancy H. C. Roosens, Wesley Mattheus.

**Investigation:** Nina Van Goethem, Dieter Van Cauteren.

**Methodology:** Nina Van Goethem, Dieter Van Cauteren.

**Project administration:** Pieter-Jan Ceyssens, Nancy H. C. Roosens, Wesley Mattheus.

**Supervision:** Dieter Van Cauteren, Wesley Mattheus.

**Visualization:** Nina Van Goethem.

**Writing – original draft:** Nina Van Goethem, Dieter Van Cauteren.

**Writing – review & editing:** An Van Den Bossche, Pieter-Jan Ceyssens, Wim Coucke, Kris Vernelen, Nancy H. C. Roosens, Sigrid C. J. De Keersmaecker, Dieter Van Cauteren, Wesley Mattheus.

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
