## [Decision Letter · Decision Letter 0]

10 Aug 2021

PONE-D-21-21282

Coverage of the national surveillance system for human Salmonella infections, Belgium, 2016-2020

PLOS ONE

Dear Dr. Van Goethem,

Thank you for submitting your manuscript to PLOS ONE. After careful consideration, we feel that it has merit but does not fully meet PLOS ONE’s publication criteria as it currently stands. Therefore, we invite you to submit a revised version of the manuscript that addresses the points raised during the review process.

Please clarify what is meant by more than 100% coverage and other points raised by the reviewers.

We look forward to receiving your revised manuscript.

Kind regards,

Iddya Karunasagar

Academic Editor

PLOS ONE

Journal Requirements:

Additional Editor Comments:

The reviewers have made very important comments. Clarification is needed on what is meant by greater than 100%. Please revise considering all the reviewer comments.

Reviewers' comments:

Reviewer's Responses to Questions

**Comments to the Author**

1. Is the manuscript technically sound, and do the data support the conclusions?

Reviewer #1: Yes

2. Has the statistical analysis been performed appropriately and rigorously? 

Reviewer #1: Yes

3. Have the authors made all data underlying the findings in their manuscript fully available?

Reviewer #1: No

4. Is the manuscript presented in an intelligible fashion and written in standard English?

Reviewer #1: Yes

5. Review Comments to the Author

Reviewer #1: Can you explain, and perhaps provide an example in the methods, for how a laboratory can have >100% coverage? [see this in the discussion - wonder if helpful to mention in the methods or as you report that some did - can simply say parenthetically something like.

XX labs had coverage greater than 100% (because they reported fewer cases in the survey than the number reported to NRC).]

It is said that you removed cases with missing info (line 230) but those should still be included in the n_1 and n_2 shouldn't they? Or at least you should provide what the estimated coverage is when you include them as a lower bound of coverage.

Seems that would be 15350 cases with coverage of 80%. It is still very high.

The data availability statement should be written in the format that the journal requests. There are clearly issues of personal identifiability, etc., and it should be clearer where to get the data.

Overall the text is written in excellent English but there are several small issues and I doubt that I capture them all here. The paper could benefit from a reading by a native English speaking editor.

Line 69 - suggest "voluntarily" vs. "voluntary"

Line 101 - suggest "previously" vs. "previous"

Line 114 - suggest "ranges" vs. "ranged" and if inculsive of 2016 and 2020 would suggest from 2016 to 2020

Line 121 - is capitalization all correct?

Line 126 - from a patient *who was* part of

Line 172 - Number *of* Salmonella

Line 177 - suggest "annual" vs. "annuals"

Line 196 - "send" instead of "sent"

Line 221 - suggest "indicated they used culture" there and elsewhere as appropriate

Line 224 - suggest "except one indicated they sent all"

Line 257 - unsure of use of "towards" here

Line 279-280 - and allowed *us* to confidently estimate

Line 289 - suggest "compared" vs. "compare"

6. PLOS authors have the option to publish the peer review history of their article (what does this mean?). If published, this will include your full peer review and any attached files.

Reviewer #1: No

---

## [Author Response · Author response to Decision Letter 0]

16 Aug 2021

Authors’ response to reviews

Title: Coverage of the national surveillance system for human Salmonella infections, Belgium, 2016-2020

Authors:

N. Van Goethem 

A. Van Den Bossche

P.-J. Ceyssens

A. Lajot

W. Coucke

K. Vernelen

N.H.C. Roosens

S.C.J. De Keersmaecker

D. Van Cauteren

W. Mattheus

Version : 1 Date : 16 August 2021

Authors' response: Thank you very much for considering our manuscript. We thank the reviewer and editorial staff for their valuable review. We carefully went through the constructive reviews provided by the peer reviewer and editor and have revised the manuscript accordingly. We have outlined our responses point by point to the reviewer’s comments. The line numbers presented here correspond to those in the final version (with track changes) of the manuscript. 

 

Reviewer 1: Can you explain, and perhaps provide an example in the methods, for how a laboratory can have >100% coverage? [see this in the discussion - wonder if helpful to mention in the methods or as you report that some did - can simply say parenthetically something like.

XX labs had coverage greater than 100% (because they reported fewer cases in the survey than the number reported to NRC).]

Authors’ response: We thank the reviewer for this comment. In addition to the clarifications that were already provided in the discussion section (L304-309), we have added the following clarifications to the methods section (L150-152): “A laboratory can have >100% coverage when they report fewer cases in the survey than the number of isolates from unique cases sent to the NRC.” Accordingly, we added the following to the results section as suggested (L226-227): “A total of 18 laboratories had a coverage greater than 100% (because they reported fewer cases in the survey than the number reported to the NRC).” 

Reviewer 1: It is said that you removed cases with missing info (line 230) but those should still be included in the n_1 and n_2 shouldn't they? Or at least you should provide what the estimated coverage is when you include them as a lower bound of coverage.

Seems that would be 15350 cases with coverage of 80%. It is still very high.

Authors’ response: We thank the reviewer for this comment. As we do not know whether there is a match between the NRC and SNL database for the cases with missing info, we did exclude them from the main analysis. However, we do agree that it might provide a ‘lower bound’ of coverage. Therefore, we have added the following to the results section (L251-254): “When taking into account the cases with missing date of birth, gender, or postal code, the estimated total number of cases was 13,350 (95% CI: 15,172-15,528) and the coverage of the NRC and SNL network were estimated to be 80% (12,316/15,350) (95% CI: 80-81%) and 31% (4,822/15,350) (95% CI: 31-32%), respectively.”

Reviewer 1: The data availability statement should be written in the format that the journal requests. There are clearly issues of personal identifiability, etc., and it should be clearer where to get the data.

Authors’ response: We thank the reviewer for this remark. We have now included all relevant data underlying the results in the Supporting information. The underlying data on Salmonella epidemiology in Belgium as presented in Fig 1 and Fig 2, is contained in S1 Table and S2 Table, respectively. The laboratory survey results aggregated per laboratory (anonymized), including the number of Salmonella cases reported in the survey, the number of Salmonella isolates from unique cases sent to the National Reference Center (NRC), the method of diagnosis used, the reasons for sending isolates to the NRC, and the serotypes sent to the NRC, are presented in S3 Table. The stool reimbursement data, used to calculate the positivity rate and to estimate the number of Salmonella diagnoses for laboratories that did not participate in the survey, are presented per RIZIV number (anonymized) in S4 Table. Finally, the numbers per year underlying the capture-recapture analyses are presented in S5 Table.

Reviewer 1: Overall the text is written in excellent English but there are several small issues and I doubt that I capture them all here. The paper could benefit from a reading by a native English speaking editor.

Line 69 - suggest "voluntarily" vs. "voluntary"

Line 101 - suggest "previously" vs. "previous"

Line 114 - suggest "ranges" vs. "ranged" and if inculsive of 2016 and 2020 would suggest from 2016 to 2020

Line 121 - is capitalization all correct?

Line 126 - from a patient *who was* part of

Line 172 - Number *of* Salmonella

Line 177 - suggest "annual" vs. "annuals"

Line 196 - "send" instead of "sent"

Line 221 - suggest "indicated they used culture" there and elsewhere as appropriate

Line 224 - suggest "except one indicated they sent all"

Line 257 - unsure of use of "towards" here

Line 279-280 - and allowed *us* to confidently estimate

Line 289 - suggest "compared" vs. "compare"

Authors’ response: We thank the reviewer for these kind words and the valuable suggestions and corrections. We have adapted all suggestions and corrections in the manuscript accordingly.

---

## [Decision Letter · Decision Letter 1]

17 Aug 2021

Coverage of the national surveillance system for human Salmonella infections, Belgium, 2016-2020

PONE-D-21-21282R1

Dear Dr. Van Goethem,

We’re pleased to inform you that your manuscript has been judged scientifically suitable for publication and will be formally accepted for publication once it meets all outstanding technical requirements.

Kind regards,

Iddya Karunasagar

Academic Editor

PLOS ONE

Additional Editor Comments (optional):

All reviewer comments have been addressed.

Reviewers' comments:

Reviewer's Responses to Questions

**Comments to the Author**

1. If the authors have adequately addressed your comments raised in a previous round of review and you feel that this manuscript is now acceptable for publication, you may indicate that here to bypass the “Comments to the Author” section, enter your conflict of interest statement in the “Confidential to Editor” section, and submit your "Accept" recommendation.

Reviewer #1: All comments have been addressed

2. Is the manuscript technically sound, and do the data support the conclusions?

Reviewer #1: Yes

3. Has the statistical analysis been performed appropriately and rigorously? 

Reviewer #1: Yes

4. Have the authors made all data underlying the findings in their manuscript fully available?

Reviewer #1: Yes

5. Is the manuscript presented in an intelligible fashion and written in standard English?

Reviewer #1: Yes

6. Review Comments to the Author

Reviewer #1: (No Response)

7. PLOS authors have the option to publish the peer review history of their article (what does this mean?). If published, this will include your full peer review and any attached files.

Reviewer #1: No

---

## [Editor Report · Acceptance letter]

19 Aug 2021

PONE-D-21-21282R1 

Coverage of the national surveillance system for human *Salmonella* infections, Belgium, 2016-2020 

Dear Dr. Van Goethem:

I'm pleased to inform you that your manuscript has been deemed suitable for publication in PLOS ONE. Congratulations! Your manuscript is now with our production department. 

Kind regards, 

on behalf of

Dr. Iddya Karunasagar 

Academic Editor

PLOS ONE